# Towards Mutation-Specific Precision Medicine in Atypical Clinical Phenotypes of Inherited Arrhythmia Syndromes

**DOI:** 10.3390/ijms22083930

**Published:** 2021-04-10

**Authors:** Tadashi Nakajima, Shuntaro Tamura, Masahiko Kurabayashi, Yoshiaki Kaneko

**Affiliations:** Department of Cardiovascular Medicine, Gunma University Graduate School of Medicine, Maebashi 371-8511, Gunma, Japan; bright_tiger_no1@yahoo.co.jp (S.T.); mkuraba@gunma-u.ac.jp (M.K.); kanekoy@gunma-u.ac.jp (Y.K.)

**Keywords:** atrial fibrillation, atypical clinical phenotype, Brugada syndrome, early repolarization syndrome, long QT syndrome, mutation, precision medicine

## Abstract

Most causal genes for inherited arrhythmia syndromes (IASs) encode cardiac ion channel-related proteins. Genotype-phenotype studies and functional analyses of mutant genes, using heterologous expression systems and animal models, have revealed the pathophysiology of IASs and enabled, in part, the establishment of causal gene-specific precision medicine. Additionally, the utilization of induced pluripotent stem cell (iPSC) technology have provided further insights into the pathophysiology of IASs and novel promising therapeutic strategies, especially in long QT syndrome. It is now known that there are atypical clinical phenotypes of IASs associated with specific mutations that have unique electrophysiological properties, which raises a possibility of mutation-specific precision medicine. In particular, patients with Brugada syndrome harboring an *SCN5A* R1632C mutation exhibit exercise-induced cardiac events, which may be caused by a marked activity-dependent loss of R1632C-Nav1.5 availability due to a marked delay of recovery from inactivation. This suggests that the use of isoproterenol should be avoided. Conversely, the efficacy of β-blocker needs to be examined. Patients harboring a *KCND3* V392I mutation exhibit both cardiac (early repolarization syndrome and paroxysmal atrial fibrillation) and cerebral (epilepsy) phenotypes, which may be associated with a unique mixed electrophysiological property of V392I-Kv4.3. Since the epileptic phenotype appears to manifest prior to cardiac events in this mutation carrier, identifying *KCND3* mutations in patients with epilepsy and providing optimal therapy will help prevent sudden unexpected death in epilepsy. Further studies using the iPSC technology may provide novel insights into the pathophysiology of atypical clinical phenotypes of IASs and the development of mutation-specific precision medicine.

## 1. Introduction

Recent advances in molecular genetics have identified many causal genes for inherited arrhythmia syndromes (IASs) such as long QT syndrome (LQTS) [1], short QT syndrome (SQTS) [2], Brugada syndrome (BrS) [3,4] and early repolarization (ER) syndrome (ERS) [3,5]. Most causal genes for IASs encode cardiac ion channels or their related proteins. Genotype-phenotype studies and functional analyses of mutant genes, using heterologous expression systems and experimental animal models, have revealed the pathophysiology of IASs and enabled the establishment of causal gene-specific precision medicine [6,7,8]. Furthermore, analyses of patient-specific and/or genome-edited induced pluripotent stem cell-derived cardiomyocytes (iPSC-CMs) have provided further insights into the pathophysiology of IASs and novel promising therapeutic strategies for IASs, although there are still some limitations of using iPSC-CMs, such as immature structure and function and mixed population of atrial, ventricular, and nodal cells, as a standard technology [9].

The altered function of causal genes that encode cardiac ion channels is caused by multiple mechanisms, including trafficking defects, producing non-functional channels, altered channel gating properties, and a combination thereof. These altered functions of mutant channels underly the clinical phenotypes of IASs [10,11,12]. Particularly, unique electrophysiological properties of mutant channels have been shown to be associated with the atypical clinical phenotypes of IASs [10,13]. Furthermore, the elucidation of the mechanisms underlying the atypical clinical phenotypes of IASs has raised the possibility of mutation-specific precision medicine.

We herein review the current knowledge of genotype-phenotype relationships, underlying molecular and cellular mechanisms, and established pharmacological therapies of IASs, including LQTS, SQTS, and J wave syndrome (BrS and ERS). Furthermore, we describe the atypical clinical phenotypes of IASs attributable to unique electrophysiological properties of mutant channels and the potential mutation-specific precision medicine.

## 2. LQTS and SQTS

### 2.1. LQTS

LQTS is a relatively prevalent inherited disorder characterized by an abnormally prolonged QT interval on an electrocardiogram (ECG) and an increased risk of polymorphic ventricular tachycardia (VT), torsade de pointes (TdP), and ventricular fibrillation (VF), leading to syncope or sudden cardiac death (SCD) [14,15]. At least 17 genes have been reported to be causal for LQTS (LQT1-17) (Table 1) [16,17,18,19,20,21,22,23,24,25,26,27,28,29,30,31]. However, reappraisal of these genes by Adler et al. has shown that more than half have limited or disputed evidence to support their disease causation [32]. Although *KCNQ1*, *KCNH2*, *SCN5A*, *KCNJ2*, *CACNA1C*, *CALM1*, *CALM2,* and *CALM3* genes are classified as definitely causal for LQTS [32], further studies are necessary to conclude the causality of the other genes for LQTS. Notably, it is also known that a part of LQTS, SQTS, BrS, and ERS share common genetic backgrounds and that some causal genes are also associated with other cardiac phenotypes, such as atrial fibrillation (AF) [33,34,35], and/or extra-cardiac phenotypes as shown in Table 1.

A loss-of-function of outward currents and/or a gain-of-function of inward currents during the plateau phase of ventricular action potential (AP) cause prolongation of AP duration (APD), leading to prolongation of QT interval on an ECG. Mutations in genes that encode these channels/currents have been causal for LQTS (Table 1) [1,8]. Among them, mutations in the first three identified genes—*KCNQ1* for LQT1, *KCNH2* for LQT2, and *SCN5A* for LQT3—account for approximately 90% of genetically affected LQTS patients, while those in other causal genes have rarely been identified [1,8].

The clinical features differ among LQT1, LQT2, and LQT3. Cardiac events are more often associated with physical activity (β-adrenergic stimulation) in LQT1 than in LQT2 or LQT3, and those in LQT3 typically occur during sleep or at rest [36]. Regarding pharmacological therapies, β-blocker therapy has been established in LQT1 and LQT2 but remains controversial in LQT3 [37], although a recent international multicenter study reported the efficacy of β-blocker therapy in patients with LQT3, especially female patients with QTc > 500 ms at baseline [38]. These apparent clinical differences between LQT1 and LQT3 can be explained by the altered cellular electrophysiology of each mutant channel.

*KCNQ1* encodes the α-subunits of slowly activating delayed rectifier potassium currents (I_Ks_) [39,40], and *SCN5A* encodes Nav1.5, the α-subunits of voltage-gated sodium channels/currents (I_Na_) [41]. The modulation of each channel function by β-adrenergic stimulation is known to differ: The *KCNQ1*-encoding I_Ks_ is markedly augmented by β-adrenergic stimulation [42], while the *SCN5A*-encoding I_Na_ appears to be less modulated by β-adrenergic stimulation than I_Ks_ [43,44,45]. Therefore, an augmentation of I_Ks_ by β-adrenergic stimulation is impaired in LQT1 patients, while it is maintained in LQT3 patients. These findings explain why cardiac events in patients with LQT1 are associated with physical activity and why β-blocker therapy has been established in LQT1 but not in LQT3. However, our own group and Chen et al. reported LQT3 patients (with a *SCN5A* V1667I or V2016M mutation) exhibiting epinephrine-induced marked QT prolongation [11,46]. Electrophysiological experiments revealed that both mutant channels caused a gain-of-function by β-adrenergic stimulation. In addition, a functional analysis of patient-specific iPSC-CMs revealed that propranolol inhibited the late sodium currents observed in N1774D-Nav1.5 [47]. Therefore, these findings may provide the rationale supporting the efficacy of β-blocker therapy in some cases of LQT3. In contrast, mexiletine, which blocks late I_Na_, is effective in LQT3 but not in LQT1 [6,8]. Intriguingly, an analysis of LQT1 patient-specific iPSC-CMs revealed that a small molecule ML277 could restore I_Ks_ function, which raised a possibility that ML277 can be a therapeutic candidate for LQT1 patients [48].

In LQT2, emotional stress and sudden auditory stimuli are major triggers for cardiac events [49,50], which might be explained by the finding that *KCNH2*, encoding the α-subunits of rapidly activating delayed rectifier potassium currents (I_Kr_), is expressed in chromaffin cells and regulates catecholamine release and that blocking KCNH2 channels leads to hyperexcitability and an increase in catecholamine release [51]. It is also known that female patients with LQT2 have an increased risk of cardiac events during the postpartum period, although the mechanisms have not yet been fully elucidated [52]. Therefore, β-blocker therapy should not be discontinued during this period. Analysis of LQT2 patient-specific iPSC-CMs revealed that lumacaftor could rescue the pathological phenotype of LQT2, particularly with *KCNH2* mutations that cause a loss-of-function through reduction of the intracellular transport or trafficking of KCNH2 proteins to the cell membrane [53].

Furthermore, a mutation-specific atypical clinical phenotype has also been reported. Clinical and experimental data have shown that the APD or QT interval on an ECG is usually prolonged during hypothermia/low temperature, while shortened during fever/high temperature [54,55]. However, some *KCNH2* mutations (A558P, G584S, and T613M) have been reported to be associated with the prolongation of the QT interval and the development of TdP during fever [8,54]. Notably, these mutations are located at the S5-pore region of KCNH2 that is related to the inactivation of the channels [56]. Amin et al. reported the mechanism whereby these mutations are associated with fever-induced QT prolongation [54]. Wild-type (WT) KCNH2 currents increased when the temperature was elevated, whereas mutant (A558P and F640V) KCNH2 currents did not markedly increase compared with WT KCNH2 currents, possibly due to their faster inactivation rate at an elevated temperature. This suggests that the lack of an increase in mutant KCNH2 currents during fever might lead to QT prolongation and the development of TdP. These findings underscore the importance of reducing the body temperature by the timely use of antipyretics in patients with mutations at the S5-pore region of KCNH2.

LQT7 and LQT8 have initially been thought to have extra-cardiac phenotypes and have been defined as Andersen-Tawil syndrome (ATS) and Timothy syndrome (TS), respectively. ATS is characterized by periodic paralysis, cardiac arrhythmias: Frequent premature ventricular contraction, bidirectional or polymorphic VT, with QT or QU prolongation, and dysmorphic features [22]. However, approximately 30% of *KCNJ2* mutation carriers exhibit only the cardiac phenotype [57]. *KCNJ2* encodes inward rectifier potassium currents (I_K1_), and a loss-of-function of I_K1_ by *KCNJ2* mutations is associated with ATS [22,58]. Regarding pharmacological therapies for ATS, β-blockers and L-type calcium channel antagonists have been used to treat ventricular tachyarrhythmias (VTAs) [59,60]. Furthermore, Miyamoto et al. reported that oral flecainide therapy is an effective and safe means of suppressing VTAs [61].

TS is characterized by multiorgan dysfunction including lethal arrhythmias, webbing of fingers and toes, congenital heart disease, immune deficiency, intermittent hypoglycemia, cognitive abnormalities, and autism [23]. *CACNA1C* encodes Cav1.2, the α-subunits of voltage-gated L-type calcium channels/currents (I_Ca_), and a gain-of-function of Cav1.2 is associated with TS [23]. However, *CACNA1C* gain-of-function mutations have been identified in patients with LQTS without extra-cardiac phenotypes (non-syndromic LQTS) [62,63]. It is noteworthy that *CACNA1C* gain-of-function mutations (G406R and G402S) that cause TS are located at distinct sites, exon 8 or exon 8a, of Cav1.2, while those that cause non-syndromic LQTS spread throughout the Cav1.2, and that non-syndromic LQTS is the common phenotype of *CACNA1C* gain-of-function mutations [23,62,63,64]. In addition to germline mutations, mosaicism also contributes to phenotypic manifestations [63]. Due to the rarity and high mortality of TS at a relatively young age, no valid drug therapy has yet been established. Verapamil, an L-type calcium channel antagonist, decreased VTAs in a patient with TS, but failed to completely eliminate VTAs and shorten the QTc interval [65]. Ranolazine, a multi potent ion-channel blocker, was even more effective in suppressing VTAs in the patient when it was added to verapamil [66,67]. Mexiletine has been reported to shorten the QTc interval through the inhibition of late I_Na_ [68]. Intriguingly, an analysis of TS patient-specific iPSC-CMs revealed that roscovitine, a cyclin-dependent kinase (CDK) inhibitor, could rescue the TS phenotypes through in part inhibiting CDK5, which provides insights into the regulation of I_Ca_ and the development of future therapeutics for TS patients [69].

Calmodulin (CAM) is encoded by three distinct genes, *CALM1-3*. Regarding CAM mutations, patient-specific iPSC-CMs could recapitulate a disruption of Ca^2+^/CAM-dependent inactivation of L-type I_Ca_, and revealed that CAM mutation-induced repolarization abnormalities could be reversed by verapamil [70,71]. Furthermore, mutant allele-specific knockout using a clustered regularly interspaced short palindromic repeats (CRISPR)-CRISPR associated protein 9 (CRISPR-Cas9) system could rescue the electrophysiological abnormalities of a *CALM2* mutation [70]. Such a latest genome-editing technology may provide a promising therapeutic approach for IASs.

Although many pathogenic mutations for LQTS have been identified, approximately 20% of LQTS remain genetically elusive. In contrast, many variants of uncertain significance (VUS) have been identified. Therefore, technologies that can determine their pathophysiological roles are desired.

Mutations in ion channel-related genes have also been identified in patients with acquired LQTS, including drug-induced LQTS (diLQTS), whereas the functional changes of mutations identified in diLQTS patients were mild compared with those in LQTS patients [72,73]. However, most diLQTS remain genetically elusive, whereas many VUS have been identified. It also remains elusive why certain individuals have a higher propensity to develop QT interval and consequently life-threatening arrhythmias in response to drugs. An analysis of diLQTS patient-specific iPSC-CMs and application of genome-editing technologies can reveal the mechanisms of diLQTS, predict the arrhythmogenic risk of drugs, and determine the pathophysiology of VUS [74]. Genome-edited iPSC-CMs harboring distinct *KCNH2* mutations (A561T within the pore region and N996I within the tail region) in momolayer cultures displayed prolonged repolarization in the pore mutation larger than tail mutation, and blocking the KCNH2 channels conferred greater susceptibility to arrhythmic events in the pore mutation compared with tail mutation [75]. These findings indicate that evaluating the drug sensitivity of patient-specific iPSC-CMs may facilitate patient risk stratification.

### 2.2. SQTS

SQTS is a very rare inherited disorder characterized by an abnormally short QT interval on an ECG and an increased risk of life-threatening arrhythmias [2,76]. SQTS is theoretically caused by an increase of the outward currents or a decrease of the inward currents during the plateau phase of ventricular AP. Indeed, a gain-of-function of outward potassium currents, such as I_Kr_, I_Ks_, and I_K1_, and a loss-of-function of inward currents, such as I_Ca_, have been associated with SQTS (Table 1) [77,78,79,80,81]. Notably, most of causal genes (*KCNQ1*, *KCNH2*, *KCNJ2*, *CACNA1C*, *CACNB2,* and *CACNA2D1*) for SQTS are ion channel-related genes that are also causal for LQTS (Table 1). Furthermore, I_Ca_-related genes (*CACNA1C*, *CACNB2,* and *CACNA2D1*) are also causal for BrS and ERS (Table 1), which is consistent with the finding that a short QT interval is often accompanied by BrS and/or ER pattern [80,82]. The gain- or loss-of-function of the same gene, *CACNA1C*, is associated with different disease entities (either TS/LQTS or SQTS, BrS and ERS). In addition to TS, I_Ca_-related genes are associated with extra-cardiac phenotypes such as autism spectrum disorder (ASD) [83]. Intriguingly, a *CANCA1C* loss-of-function mutation (K800T) causes a new Cav1.2 channelopathy with a short QT interval and extra-cardiac phenotypes such as ASD and severe dental enamel defects [84].

*KCNH2* is the major causal gene, but its mutations account for only approximately 15% of SQTS patients [2]. Mutations in *KCNQ1* and *KCNJ2* have been less frequently identified, and those of the remaining genes have been rarely identified. Risk stratification and valid pharmacological therapies have not yet been established due to the low number of patients with SQTS [2]. Quinidine, which blocks several potassium channels including I_to_ and I_Kr_, was reported to prolong the QT interval and reduce the incidence of life-threatening arrhythmias in patients with SQTS [85,86]. Other antiarrhythmic drugs, such as sotalol, amiodarone, disopyramide, and β-blockers, failed to show beneficial effects on SQTS [86].

## 3. J Wave Syndrome (BrS and ERS)

### 3.1. BrS

BrS is characterized by coved-type ST segment elevations in the right precordial ECG leads with or without drug (sodium channel blocker) provocation, and is associated with fatal arrhythmias leading to syncope or SCD [8,87]. Cardiac events of BrS typically begin to occur in adolescence. Notably, cardiac events occur more often during sleep or at rest than during vigorous physical activity [88].

To date, more than 20 genes have been reported to be related with BrS (Table 1). *SCN5A* was first identified as a causal gene for BrS [89]. Later, over 20 other genes were identified as causal or modifier genes for BrS [80,82,90,91,92,93,94,95,96,97,98,99,100,101,102,103,104,105,106,107,108]. Mutations in *SCN5A* account for approximately 20% of BrS cases, while those of other genes rarely do [8]. However, among the genes reported to be causal or modifier for BrS, reappraisal of these genes by Hosseini et al. suggested that only *SCN5A* had definite evidence of being a causal gene [109]. Therefore, further studies are required to conclude that these reported genes are indeed causal for BrS.

Regarding the cellular mechanisms underlying BrS, there are two hypotheses: The repolarization hypothesis and the depolarization hypothesis, which are still in debate [3]. Among those, however, the repolarization hypothesis appears to be supported from the viewpoint of the abnormal functions of causal genes.

Transient outward potassium currents (I_to_) are more abundantly expressed in the right ventricle (RV) than in the left ventricle (LV) and at the epicardium than at the endocardium, which contributes to the formation of the phase 1 notch and “spike and dome” morphology at RV epicardium (Figure 1A) [110,111]. The predominance of outward currents over inward currents at the epicardium in RV outflow tract (RVOT) during the early phase of ventricular APs, due to a loss-of-function of inward currents (such as I_Na_ and I_Ca_) or a gain-of-function of outward currents (such as I_to_ and ATP-sensitive inward rectifier potassium currents [I_K-ATP_]), can augment the AP notch (Figure 1A), thereby inducing coved-type ST segment elevations in the right precordial ECG leads (Figure 1B), and result in the development of fatal arrhythmias through a so-called phase 2 re-entry [3,112]. During the slower heart rate or lower physical activity, I_to_ is augmented due to a recovery from inactivation and I_Ca_ is decreased due to a reduced β-adrenergic stimulation, while I_Na_ is minimally affected (Figure 2) [11,113,114]. The decreased heart rate- and reduced β-adrenergic stimulation-induced predominance of outward currents over inward currents during the early phase of RVOT APs can explain why cardiac events of BrS tend to occur during sleep or at rest [3,112]. In cases of electrical storm (ES), the use of isoproterenol, which increases I_Ca_ and decreases I_to_ due to an increase of heart rate, is recommended to suppress fatal arrhythmias [3,8]. Drugs that have an inhibitory effect on I_to_, such as quinidine and bepridil, and those that increase the heart rate and I_Ca_, such as cilostazol, have been shown to be somewhat effective as adjunctive therapy, although implantable cardioverter defibrillator is the only available therapy for preventing SCD [11,112].

The genotype-phenotype relationships for BrS have been less thoroughly clarified than those for LQTS. It has long been unclear whether or not the presence of *SCN5A* mutations is associated with the severity of BrS. However, Yamagata et al. recently reported that it can predict the severity of BrS [115]. *SCN5A* mutations can be associated with other arrhythmic and cardiomyopathic phenotypes such as sinus node dysfunction (SND), atrioventricular block, supraventricular tachyarrhythmias (SVT), LQTS, dilated cardiomyopathy and LV noncompaction (Table 1) [10,116,117,118,119,120,121]. BrS phenotype can be aggravated by a fever [122,123,124], but this phenomenon appears to be restricted to patients with *SCN5A* mutations [125]. As mentioned above, mutations in genes that encode I_Ca_ are often accompanied by ER and a relatively short QT interval (Table 1) [80,82].

While cardiac events of BrS typically occur during sleep or at rest, we encountered a familial case of BrS who exhibited exercise-induced cardiac events, an atypical clinical phenotype as BrS [10]. In addition to the BrS phenotype, the proband (a 17-year-old male) had SVT and SND, and his mother also had BrS and SND. Both carried an *SCN5A* R1632C mutation, located at domain IV-segment 4 (DIVS4).

The SCN5A/Nav1.5 is composed of four homologous but non-identical domains (DI-DIV), and each domain contains an S4 voltage sensor that consists of positively charged arginine and lysine repeats. The S4 segments in SCN5A have domain-specific functions, and the DIVS4 plays a key role in the activation and fast inactivation processes through the coupling of arginine residues in DIVS4 with residues of the putative gating charge transfer center in DIVS1-3 [126,127,128,129]. Thus, the R1632 position in SCN5A was thought to be an important position in the activation and inactivation processes. A functional analysis of the *SCN5A* R1632C mutation revealed that it displayed a marked delay of recovery from fast inactivation, in other words, an enhanced fast-inactivated state stability of Nav1.5 [10,130]. Rapid repetitive depolarizing pulses induced a marked reduction of the current amplitudes in R1632C-Nav1.5 compared with WT-Nav1.5. This means that R1632C-Nav1.5 will markedly decrease compared with WT-Nav1.5 during higher heart rate or vigorous physical activity (Figure 2). This unique electrophysiological property, a marked activity-dependent loss of Nav1.5 availability due to a marked delay of recovery from fast inactivation, might be associated with the atypical clinical phenotype of this mutation carriers. Given that isoproterenol infusion increases the heart rate, thereby decreasing R1632C-Nav1.5, the use of isoproterenol should be avoided in carriers of this mutation in cases of ES. Conversely, there is a possibility that β-blocker therapy may be effective for the suppression of ES in carriers of this mutation.

Patient-specific iPSC-CMs harboring other *SCN5A* mutations have been reported to be able to recapitulate cellular phenotypic features of BrS, such as reduced I_Na_, reduced maximal upstroke velocity of AP, increased burden of triggered activity, and abnormal Ca^2+^ transients. [9,131]. For the *SCN5A* R1632C mutation carriers, in addition to clinical studies, an analysis of iPSC-CMs may promote the development of mutation-specific precision medicine.

### 3.2. ERS

ER (or J wave) is generally defined as J-point elevation (≥0.1 mV) in ≥2 inferior and/or lateral ECG leads [3]. ER is a common ECG finding that affects 1% to 5% of people, and was long considered “benign”. However, several reports suggested its potential arrhythmogenicity [132,133], and in 2008, a multicenter study by Haissaguerre et al. revealed an obvious association of ER and arrhythmogenicity [134]. Thereafter, ER began to attract the attention of many cardiologists and researchers.

ERS is now generally diagnosed in patients who display ER in the inferior and/or lateral ECG leads presenting with aborted cardiac arrest, documented VF or polymorphic VT [3]. As with BrS, although both the repolarization and depolarization hypotheses have been proposed as the cause of ER, the repolarization hypothesis is prevailing as a cause of “malignant” ER, recognized in ERS [3]. Regarding the cellular mechanisms, a marked increase of AP notch in the epicardium compared with the endocardium in the LV underlies the generation of ER in the inferior and lateral ECG leads (Figure 1A,B). An abundance of I_to_ in the epicardium relative to the endocardium in the LV largely contributes to the formation of the AP notch in the epicardium (Figure 1A,B) [135]. In addition, the predominance of outward currents over inward currents in the LV epicardium during the early phase of ventricular AP, due to a loss-of-function of inward currents (such as I_Na_ and I_Ca_) or a gain-of-function of outward currents (such as I_to_ and I_K-ATP_), can theoretically augment the AP notch, thereby leading to the generation of an ER pattern (Figure 1A,B) [5,135].

Indeed, all causal genes for ERS that have been identified encode ion channel-related proteins and mutations in these genes cause either a loss-of-function of inward currents (I_Na_ and I_Ca_) or a gain-of-function of outward currents (I_to_ and I_K-ATP_) during the early phase of ventricular AP (Table 1) [82,95,100,136,137,138]. Of note, the genetic backgrounds of ERS share similarities with those of BrS (Table 1), although the association of ERS with these genes should also be reappraised, as with BrS. ERS with I_Ca_-related genes tends to be accompanied by BrS and/or a relatively short QT interval [82]. The clinical characteristics of ERS also share some similarities with those of BrS: Male predominance, cardiac events in adolescence, and effective pharmacological therapies such as isoproterenol, quinidine, bepridil, and cilostazol [3,139].

Among causal genes for ERS, *KCND3* mutations were recently identified [13,138]. *KCND3* encodes Kv4.3 composing transient outward potassium currents (I_to_ or I_A_), and it is expressed in both the heart and brain [140]. In the brain, *KCND3* plays a particularly important role in the development of the cerebellum [141,142], while it affects cardiac APs in both the atrium and ventricle [111]. Therefore, an altered Kv4.3 function can theoretically be associated with both cardiac and cerebral phenotypes. Indeed, a decreased Kv4.3 function caused by *KCND3* mutations has been shown to be associated with spinocerebellar ataxia (SCA)19/22 phenotype [143,144]. In contrast, an increased Kv4.3 function caused by *KCND3* mutations has been associated with cardiac phenotypes, such as BrS and AF [96,145]. Takayama et al. recently reported a 12-year-old boy with ERS associated with a *KCND3* G306A mutation [138]. In addition to the ERS phenotype, the patient had paroxysmal AF (PAF), refractory epilepsy, and intellectual disability (ID) from 2 years of age. They also reported the electrophysiological properties of the mutation: An increase of Kv4.3 (due to an increased current density and delayed inactivation) and a decrease of the increased Kv4.3 (due to a delayed recovery from inactivation) (Figure 2). Thus, an increase of Kv4.3 is thought to be associated with ERS.

We recently identified a *KCND3* V392I mutation in a familial case (mid to late teen sisters) with both cardiac (ERS, PAF) and cerebral (epilepsy, which manifested in early childhood, and ID) phenotypes [13]. Giudicessi et al. reported that the *KCND3* V392I mutation displayed a unique mixed electrophysiological property: An increase of Kv4.3 (due to an increased current density and delayed inactivation) and a decrease of the increased Kv4.3 (due to a delayed recovery from inactivation) [146], which resemble those of the *KCND3* G306A mutation [138]. As shown in Figure 2, these two mutations increase Kv4.3/I_to_ during bradycardia, thereby leading to J wave augmentation and shortening of the QTc interval, while it decreases the increased Kv4.3/I_to_ during tachycardia, thereby leading to the decrease/disappearance of the J wave and normalization of the QTc interval in both the ventricle and atrium. An increase of J wave amplitude during bradycardia may be consistent with the fact that one patient with this mutation suddenly died at midnight [13]. Regarding PAF, a shortening of the APD in the atrium during bradycardia may contribute to the occurrence of AF, possibly consistent with the fact that PAF predominantly occurs at night or in the early morning [13]. On the other hand, the mechanisms by which the *KCND3* V392I mutation causes epilepsy and ID remain unclear. An increase of Kv4.3/I_A_ in the brain may be associated with epilepsy, as with a *KCND2* V404M mutation [147]. Alternatively, it is also conceivable that a unique mixed electrophysiological property of Kv4.3/I_A_ may be associated with epilepsy, as cerebral cells may be able to become excited frequently enough to decrease Kv4.3/I_A_ due to a marked delay of recovery from inactivation.

Given that the *KCND3* G306A mutation associated with both cardiac (ERS and PAF) and cerebral (epilepsy and ID) phenotypes also displays a unique mixed electrophysiological property, we have proposed a link between *KCND3* mutations with a unique mixed electrophysiological property and cardiocerebral phenotypes, which may be defined as a novel cardiocerebral channelopathy [13]. The oral administration of quinidine appeared to be effective for ERS and PAF in patients with these *KCND3* mutations [13,138]. Since the epileptic phenotype appears to manifest prior to VTAs or PAF in patients with this novel cardiocerebral channelopathy [13,138], identifying *KCND3* mutations with a unique mixed electrophysiological property in patients with epilepsy of unknown etiology and providing optimal therapy to these patients will help prevent sudden unexpected death in epilepsy. The utilization of iPSC technology may provide further insights into the pathophysiology of this novel cardiocerebral channelopathy and novel therapeutic strategies.

**Table 1 ijms-22-03930-t001:** Cardiac and extra-cardiac phenotypes associated with causal or modifier genes for long QT syndrome (LQTS), short QT syndrome (SQTS), Brugada syndrome (BrS) [3,4], and early repolarization syndrome (ERS).

Gene	Cardiac Phenotype/Extra-Cardiac Phenotype (Function) [Reference]	Gene	Cardiac Phenotype/Extra-Cardiac Phenotype (Function) [Reference]
*ABCC9*	BrS (I_K-ATP_↑) [100], ERS (I_K-ATP_↑) [100], AF (I_K-ATP_↓) [33]	*KCNH2*	LQTS (I_Kr_↓) [17,53,75], SQTS (I_Kr_↑) [78], AF (I_Kr_↓, I_Kr_↑) [33]
*AKAP9*	LQTS (I_Ks_↓) [26]	*KCNJ2*	LQTS (I_K1_↓) [57], SQTS (I_K1_↑) [79], AF (I_K1_↑) [33]/ATS (I_K1_↓) [22,58]
*ANK2*	LQTS (NCX/NKA/IP3R↓) [19]	*KCNJ5*	LQTS (I_K-Ach_↓) [28], AF (I_K-Ach_↑) [35], SND (I_K-Ach_↑) [35]
*CACNA1C*	LQTS (I_Ca_↑) [62,63], SQTS (I_Ca_↓) [80], BrS (I_Ca_↓) [80], ERS (I_Ca_↓) [82]/TS (I_Ca_↑) [23,64,69], ASD (I_Ca_↓) [83,84]	*KCNJ8*	BrS (I_K-ATP_↑) [95], ERS (I_K-ATP_↑) [95], AF (I_K-ATP_↑) [33]
*CACNA2D1*	SQTS (I_Ca_↓) [80], BrS (I_Ca_↓) [82], ERS (I_Ca_↓) [82]	*KCNQ1*	LQTS (I_Ks_↓) [16,48], SQTS (I_Ks_↑) [77], AF (I_Ks_↑) [33]
*CACNB2*	SQTS (I_Ca_↓) [80], BrS (I_Ca_↓) [80], ERS (I_Ca_↓) [82]/ASD [83]	*PKP2*	BrS (I_Na_↓) [105]
*CALM1*	LQTS (I_Ca_↑) [29,71]	*RANGRF/MOG1*	BrS (I_Na_↓) [97]
*CALM2*	LQTS (I_Ca_↑) [29,70]	*SCN10A*	BrS (I_Na_↓) [107], AF (I_Na_↑, I_Na_↓) [33]
*CALM3*	LQTS (I_Ca_↑) [30]	*SCN1B*	BrS (I_Na_↓) [92], AF (I_Na_↓) [33]
*CAV3*	LQTS (I_Na_↑) [24]	*SCN2B*	BrS (I_Na_↓) [101], AF (I_Na_↓) [33]
*FGF12*	BrS (I_Na_↓) [102]	*SCN3B*	BrS (I_Na_↓) [94], AF (I_Na_↓) [33]
*GPD1L*	BrS (I_Na_↓) [90,91]	*SCN4B*	LQTS (I_Na_↑) [25], AF (I_Na_↓) [33]
*HCN4*	BrS (I_h_↓) [108], SND (I_h_↓) [33], AF (I_h_↓) [33]	*SCN5A*	LQTS (I_Na_↑) [18], BrS (I_Na_↓) [89,131], ERS (I_Na_↓) [136], SND (I_Na_↓) [116], AVB (I_Na_↓) [121], AF (I_Na_↑, I_Na_↓) [33], SVT (I_Na_↓) [10], DCM (ω-currents↑) [129], LVNC [120]
*HEY2*	BrS (I_Na_↓) [103]	*SEMA3A*	BrS (I_Na_↓) [106]
*KCND2*	ERS (I_to_↑) [137], AF (I_to_↑) [34]/Epilepsy (I_A_↑) [147], ASD (I_A_↑) [147]	*SLC4A3*	SQTS (AE3↓) [81]
*KCND3*	BrS (I_to_↑) [96], ERS (I_to_↑) [138], AF (I_to_↑) [145]/SCA (I_A_↓) [143,144], Epilepsy (I_A_↑↓) [13,138]	*SLMAP*	BrS (I_Na_↓) [99]
*KCNE1*	LQTS (I_Ks_↓) [20], AF (I_Ks_↑) [33]	*SNTA1*	LQTS (I_Na_↑) [27]
*KCNE2*	LQTS (I_Kr_↓) [21], AF (I_Ks_↑) [33]	*TRDN*	LQTS (ECC↓) [31]
*KCNE3*	BrS (I_to_↑) [93], AF (I_Ks_↑) [33]	*TRPM4*	BrS (I_Na_↓) [104]
*KCNE5*	BrS (I_to_↑) [98], AF (I_Ks_↑) [33]		

AE3: Anion exchanger 3; AF: Atrial fibrillation; ASD: Autism spectrum disorder; AVB: Atrioventricular block; BrS: Brugada syndrome; DCM: Dilated cardiomyopathy; ECC: Excitation-contraction coupling; ERS: Early repolarization syndrome; I_Ca_: Voltage-gated calcium channels/currents; I_h_: Hyperpolarization-activated non-selective cation channels/currents; I_K_: Delayed rectifier potassium currents; I_Kr_: Rapidly activating I_K_; I_Ks_: Slowly activating I_K_; I_K-Ach_: Acetylcholine-activated inward rectifier potassium currents; I_K-ATP_: ATP-sensitive inward rectifier potassium currents; I_K1_: Inward rectifier potassium currents; I_Na_: Voltage-gated sodium channels/currents; IP3R: Inositol 1,4,5-trisphosphate receptor; I_to_ or I_A_: Transient outward potassium currents; LQTS: Long QT syndrome; LVNC: Left ventricular noncompaction; NCX: Na/Ca exchanger; NKA: Na/K ATPase; SCA: Spinocerebellar ataxia; SND: Sinus node dysfunction; SQTS: Short QT syndrome; SVT: Supraventricular tachyarrhythmia; ↑: Gain-of-function; ↓: Loss-of-function.

## 4. Conclusions

The combination of a thorough investigation of the clinical phenotypes of IASs and elucidation of the electrophysiological properties of mutant channels can reveal the pathophysiology of IASs. Particularly, the unique electrophysiological properties of mutant channels can be responsible for the atypical clinical phenotypes of IASs, raising the possibility of mutation-specific precision medicine in IASs. Further studies using the iPSC technology may provide further insights into the pathophysiology of atypical clinical phenotypes of IASs and the development of mutation-specific precision medicine.

## Figures and Tables

**Figure 1 ijms-22-03930-f001:**
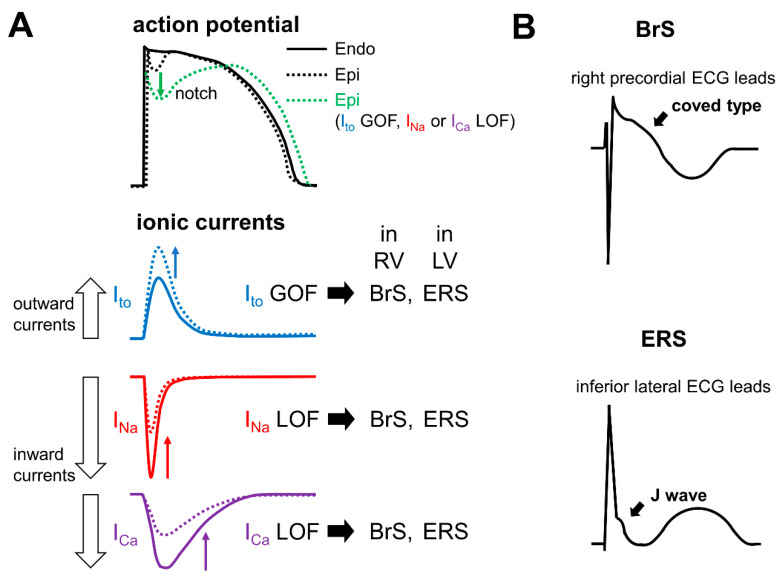
Schematic representation of the ionic mechanisms underlying Brugada syndrome (BrS) (coved-type ST segment elevations in the right precordial ECG leads) and early repolarization syndrome (ERS) (J waves in inferior and lateral ECG leads). (**A**) The upper panel shows the schematic representation of the ventricular action potentials (APs) in endocardium (solid line), epicardium with an AP notch (dotted line), and epicardium with an enhanced AP notch (green dotted line). Adapted from Antzelevitch. [112]. The lower panel shows the schematic representation of the major ionic currents that contribute to the early phase of ventricular AP. I_to_ (blue solid line), I_Na_ (red solid line), and I_Ca_ (purple solid line) are shown. The gain-of-function of I_to_ (blue dotted line), loss-of-function of I_Na_ (red dotted line) or I_Ca_ (purple dotted line) can theoretically produce an AP notch (green dotted line in (**A**)). (**B**) Schematic representation of ECGs of BrS and ERS. An enhanced AP notch at the right ventricular (RV) wall can theoretically induce coved-type ST segment elevations in the right precordial ECG leads (upper panel), while an enhanced AP notch at the left ventricular (LV) wall can theoretically produce J waves in the inferior and lateral ECG leads (lower panel). BrS: Brugada syndrome; ECG: Electrocardiogram; Endo: Endocardium; Epi: Epicardium; ERS: Early repolarization syndrome; GOF: Gain-of-function; I_Ca_: Voltage-gated L-type calcium channels/currents; I_Na_: Voltage-gated sodium channels/currents; I_to_: Transient outward potassium currents; LOF: Loss-of-function; LV: Left ventricle; RV: Right ventricle.

**Figure 2 ijms-22-03930-f002:**
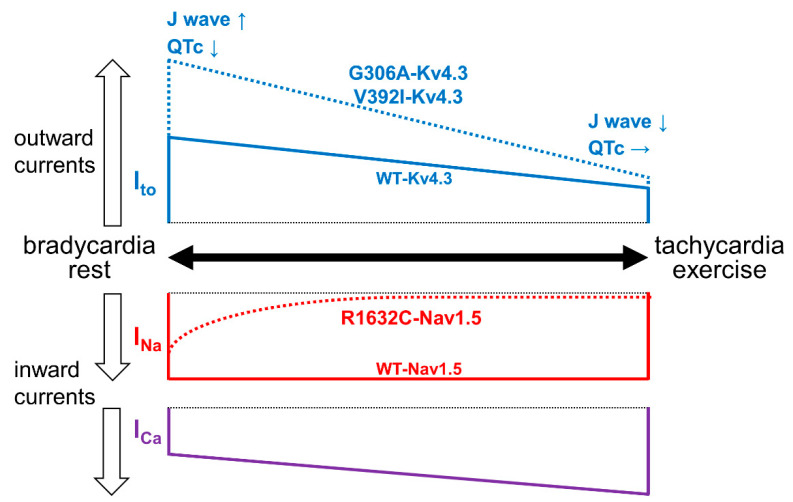
Schematic representation of the heart rate- and physical activity-dependent channel functions of the *SCN5A* R1632C mutation and *KCND3* V392I and G306A mutations. Major ionic currents, I_to_ (blue solid line), I_Na_ (red solid line), and I_Ca_ (purple solid line), during the early phase of AP are shown. Since the *SCN5A* R1632C mutation induces a marked activity-dependent loss of Nav1.5 availability due to a marked delay of recovery from fast inactivation, R1632C-Nav1.5 markedly decreases during tachycardia/exercise (red dotted line). On the other hand, since the *KCND3* V392I and G306A mutations induce an increase of current density and delayed inactivation, V392I-Kv4.3 and G306A-Kv4.3 markedly increase during bradycardia/rest (blue dotted line), while since the *KCND3* V392I and G306A mutations induce a marked delay of recovery from inactivation, V392I-Kv4.3 and G306A-Kv4.3 decrease the increased Kv4.3 during tachycardia/exercise (blue dotted line). WT: Wild-type.

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
