# Peer review of "Towards Mutation-Specific Precision Medicine in Atypical Clinical Phenotypes of Inherited Arrhythmia Syndromes"

_ijms, 2021, doi:10.3390/ijms22083930_

Round 1
Reviewer 1 Report
The authors took into considerations all remarks of the first review round. I therefor do not have any additional comments.
Reviewer 2 Report
My concerns have been adequately addressed and therefore have nothing further to add.
This manuscript is a resubmission of an earlier submission. The following is a list of the peer review reports and author responses from that submission.
Round 1
Reviewer 1 Report
The manuscript is a review of mutations in major cardiac ion channels that predispose to arrhythmias with clinical implications in patients. Overall, the manuscript is well-written and structured; however, the theme is a reiteration of current knowledge in the filed and offers no insights into strategies for future studies and how such findings are likely to advance the field. Notably there has been significant advancement in the utilization of hiPSC-CMs to further understand the molecular mechanisms of some of the major mutations, but this was not mentioned or discussed.
Table 1 states mutations in ion channels that have been comprehensively studied but the authors essentially list one reference per description and some of these references are old. The rationale for this is not clear. The table should be significantly improved to include more recent references or removed.
The theme of the manuscript should be focused on new knowledge rather than restating what a series of other reviews in the literature offer.
The authors should consider a section on drug-induced QT prolongation and contrast this with QT prolongation due to inherited mutations in ion channels.
Reviewer 2 Report
This manuscript is dealing with a very specific area of management for a relatively narrow cohort of patients with particular mutations causing inherited arrhythmia syndromes (IASs). Since most of these syndromes are caused by genetic defects in genes encoding structure and function of cardiac ion channels (or their subunits) contributing to action potential generation they are also designated channelopathies. Since the discovery of the first sentinel channelopathy-causative genes in 1995, genetic testing for channelopathies has advanced and today, the discussion about a causative role of specific mutations is taking a place. The correct identification of the definitive disease-causing mutation enables appropriate therapy. Mutation-specific genetic testing has diagnostic, prognostic, and therapeutic implications. There are more than 50 distinct channelopathy/cardiomyopathy-associated genes with hundreds of discrete missense, nonsense, insertion/deletion, frameshift, and splice site mutations.
This manuscript summarizes the current knowledge of genotype-phenotype relationships, underlying molecular and cellular mechanisms, and established pharmacological therapies in long QT syndrome (LQTS), short QT syndrome (SQTS), Brugada syndrome (BrS) and early repolarization (ER) syndrome (ERS). The main contribution is well-written and easy to read description of genetic background with consequent cardiac outward and inward currents functioning of the most encountered cardiac channelopathies. It is written in a manner potentially understandable for a broad spectrum of readers, scientists and clinicians with not only particular interest in cardiac electrophysiology. Furthermore, authors describe the atypical clinical phenotypes of the above mentioned channelopathies attributable to unique electrophysiological properties of mutant channels and suggest the potential mutation-specific management of patients.
Comments:
- Line 191. I suppose that by southeast Again you meant southeast Asia…
- Line 315. Indeed, all causal genes for ERS that have been identified encode ion channel-related proteins and mutations in these genes cause either a gain-of-function of outward currents (INa and ICa) or a loss-of-function of inward currents (Ito and IK-ATP) during the early phase of ventricular AP.
This is in contradiction with the previous sentence starting in line 310.: In addition, the predominance of outward currents over inward currents in the LV epicardium during the early phase of ventricular AP, due to a loss-of-function of inward currents (such as INa and ICa) or a gain-of-function of outward currents (such as Ito and IK-ATP), can theoretically augment the AP notch, thereby leading to the generation of an ER pattern.
Please clarify and correct !
Reviewer 3 Report
Major comments:
- From the abstract it is not clear that this article is a review. A lot of emphasis is put on the KCND3 gene while this concerns only a part (± 25%) of the main text.
- The authors state they review genotype-phenotype relationships (line 48). A table in which per gene all reported mutations with their respective phenotypes are summarized, would be an asset.
- Mutations in CACNA1C can result in Timothy syndrome or LQTS without extra-cardiac features. Can you elaborate on the genotype-phenotype correlation here?
- A different layout for both table 1 and table 2 would be appropriate. Especially in table 2, I do think the subtitles, eg. INa-related gene, are not needed since it is clear from the function column it is a INa related gene.
- The title of the paper states: "Potential mutation-specific management ... " but the authors do not go into detail on how clinical practices could be improved to establish mutation-specific management strategies.
Minor comment:
- Line 40-43: please rephrase
- Line 74-81: sentence is too long
- Line 190-192: it is not clear what you mean by this sentence